# Deep Multiple Instance Learning for Taxonomic Classification of Metagenomic Read sets

## Abstract

Metagenomic studies have increasingly utilized sequencing technologies in order to analyze DNA fragments found in environmental samples. It can provide useful insights for studying the interactions between hosts and microbes (Methé et al., 2012; Qin et al., 2010), infectious disease proliferation (Chiu & Miller, 2019), and novel species discovery (Nayfach et al., 2019). One important step in this analysis is the taxonomic classification of those DNA fragments. Of particular interest is the determination of the distribution of the taxa of microbes in metagenomic samples. Recent attempts using deep learning focus on architectures that classify single DNA reads independently from each other. In this work, we attempt to solve the task of directly predicting the distribution over the taxa of whole metagenomic read sets. We formulate this task as a Multiple Instance Learning (MIL) problem. We extend architectures used in single-read taxonomic classification with two different types of permutation-invariant MIL pooling layers: a) deepsets and b) attention-based pooling. We illustrate that our architecture can exploit the co-occurrence of species in metagenomic read sets and outperforms the single-read architectures in predicting the distribution over the taxa at higher taxonomic ranks.

## 1 Introduction

Over the last decades, advancements in sequencing technology have led to a rapid decrease of the cost of genome sequencing (Wetterstrand, 2013) while the amount of sequencing data being generated has vastly increased. This is attributable to the fact that genome sequencing is a tool of utmost importance for a variety of fields, such as biology and medicine, where it is used to identify changes in genes or aid in the discovery of potential drugs (Methé et al., 2012; Qin et al., 2010). Metagenomics is a subfield of biology, which is concerned with the study of genetic material found in samples taken directly from the environment (Consortium et al., 2016; Howe et al., 2014). DNA fragments found in those samples can be sequenced using various sequencing technologies, such as Illumina, PacBio, and Oxford Nanopore (Quail et al., 2012). This process results in substrings sampled from random positions in the genomes of the organisms, called DNA *reads*. The reads obtained from sequencing are noisy, meaning that some of the letters (called *base pairs*) are flipped to a different letter or, in some cases, the deletion or insertion of additional base pairs can occur. The error rate and the distribution of the noise is dependent on the technology used to sequence the DNA fragments (Quail et al., 2012). Newer long-read technologies can sequence complete genomes of viruses and small bacteria, but with a higher error rate (Jain et al., 2016).

As an application of metagenomic sequencing, samples can be taken from the human intestine in order to characterize the microbial flora of the human gut (Methé et al., 2012; Qin et al., 2010). Significant efforts have been carried out by projects such as the Human Microbiome Project (HMP) (Methé et al., 2012) and the Metagenomics of the Human Intestinal Tract (MetaHIT) project (Qin et al., 2010) in order to understand how the human microbiome can have an effect on human health. An important step in this process is to classify DNA fragments into various groups at different taxonomic ranks. The NCBI Taxonomy maintains a tree ontology of taxonomic labels (Wheeler et al., 2006). Organisms are assigned taxonomic labels and thus are placed on the tree. Each level of the tree represents a different taxonomic rank, with finer ranks such as *species* and *genus* being close to the leaf nodes and coarser ranks such as *phylum* and *class* closer to the root.

One approach that has shown great promise for biological classification tasks is deep learning. In recent years, we have seen various attempts of using deep learning to solve tasks such as variant calling (Poplin et al., 2018) or the discovery of DNA-binding motifs (Zou et al., 2018). These methods even outperform more classical approaches, despite the relative lack of biological prior knowledge incorporated into those models.

We consider the problem of metagenomic classification, where each individual read is assigned to a label or multiple labels corresponding to its taxon at each taxonomic rank. One could simply identify the taxon at the finest level of the taxonomy and then extract the taxa at all levels of the tree above it by following the path to the root. The problem with this approach is that for certain reads, we might not be able to accurately identify the species of the host organism, but nevertheless be interested in coarser taxonomic ranks. This can apply in cases where little relevant reference data is available for a sequencing dataset (such as deep sea metagenomics data (Tully et al., 2018) or New York City metagenomics where only $48\%$ of samples matched a known species (Afshinnekoo et al., 2015)), so a more accurate prediction at higher taxonomic ranks may be more informative for downstream analysis (Rojas-Carulla et al., 2019). Furthermore, in many cases we are only interested in the distribution of organisms in an environmental sample, also known as the *microbiota*, rather than in the classification of individual fragments.

We formulate this task as an instance of Multiple Instance Learning (MIL). MIL is a specific framework of supervised learning approaches. In contrast to the traditional supervised learning task, where the goal is to predict a value or class for each sample, in MIL, given a set of samples, the goal is to assign a value to the whole set. A set of items is called a *bag*, whereas each individual item in the bag is called an *instance*. In other words, a bag of instances is considered to be one data point (Foulds & Frank, 2010). More formally, a bag is a function $\mathbf{B} : \mathcal{X} \to \mathbb{N}$ where $\mathcal{X}$ is the space of instances. Given an instance $x \in \mathcal{X}$, $\mathbf{B}(x)$ counts the number of occurrences of $x$ in the bag $\mathbf{B}$. Let $\mathcal{B}$ be the class of such bag functions. Then the goal of a MIL model is to learn a bag-level concept $c : \mathcal{B} \to \mathcal{Y}$ where $\mathcal{Y}$ is the space of our target variable.

In the context of metagenomic classification, we consider the instances to be DNA reads. Our goal is to directly predict the distribution over a given set of taxonomic ranks in the read set (the bag). So for each taxon, our output is a real number in $[0, 1]$ denoting the portion of the reads in the read set that originated from that particular species. The motivation for this is that in a realistic set of reads, closely related organisms tend to appear together. It might thus be possible to exploit the co-occurrence of organisms to gain better accuracy (Carbonneau et al., 2018).

Our main contributions are:

- A new method to generate synthetic read sets with realistic co-occurrence patterns from collections of reference genomes.

- A novel machine learning model for predicting the distribution over taxa in a read set, combining state-of-the-art deep DNA classification models with read-set-level aggregation in a multiple instance learning setting.

- A thorough empirical assessment of our proposed model, showing superior performance in predicting the distributions of higher level taxa from read sets.

In the rest of this paper, we give an overview of previous related work in Section 2, describe our data generation method and machine learning models in Section 3 and analyse the results of our experiments in Section 4. An overview of our proposed architectures is depicted in Figure 1.

## 2 RELATED WORK

To solve the problem of metagenomic classification, more traditional methods rely on read alignment to classify each DNA fragment. Given a DNA read, one first needs to match $k$-mers to a large database of reference genomes. This is done to detect candidate segments of the genomes and can be executed quickly by first creating an index of the reference genomes during a preprocessing phase (Altschul et al., 1990; Bowe et al., 2012; Muggli et al., 2017). Following this step, one needs to use approximate string matching techniques to match the string to the candidate segments determined by the $k$-mer matching step before. A well-known and widely used tool that uses alignment is BLAST,

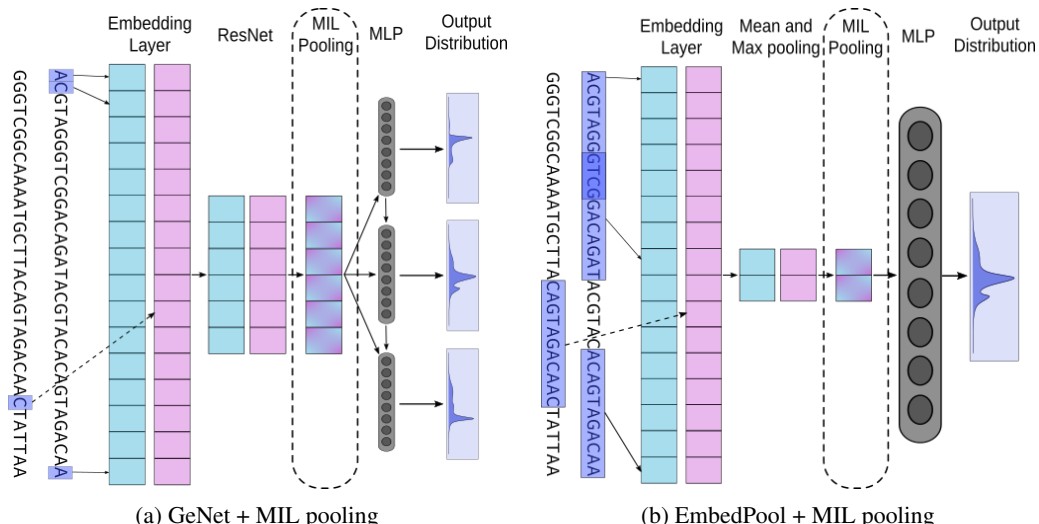

(a) GeNet + MIL pooling    (b) EmbedPool + MIL pooling

Figure 1: The two proposed architectures for solving the MIL task. The models can process multiple reads (only two reads shown for compactness) independently from each other. During the *MIL pooling* phase, the outputs for each read are combined to create a representation for the whole read set. Subsequently, the model can use this to directly predict the distribution over the taxa.

which is a general heuristic tool for aligning genomic sequences. Other alignment and mapping based tools specifically designed for metagenomics include Centrifuge (Kim et al., 2016), Kraken (Wood & Salzberg, 2014), MetaPhlAn (Segata et al., 2012), and MEGAN (Huson et al., 2007). These methods make trade-offs of sensitivity for scalability. For example, BLAST is highly sensitive, but not scalable to databases of unassembled sequencing data, while more approximate methods like Kraken are well suited for such large databases. Moreover, recent deep learning approaches have outperformed these methods by significant margins, especially in high error-rate settings (Rojas-Carulla et al., 2019; Liang et al., 2019).

Most of the previous attempts using machine learning focused on 16S rRNA sequences due to their high sequence conservation across a wide range of species. An example is the RDP (Ribosomal Database Project) classifier which uses a Naive Bayes classifier to classify 16S rRNA sequences (Wang et al., 2007). The disadvantage of this method is the loss of positional information due to the encoding of the sequence as a 'bag' of 8-letter words. However, the generalizability of this model to sequencing data drawn from other genomic regions is unclear. Similarly, La Rosa et al. (2015) use probabilistic topic modeling was used in order to classify 16S rRNA sequences in the taxonomic ranks from phylum to family. Another interesting approach is taken by Brady & Salzberg (2009) which uses Markov models to classify DNA reads and can even be combined with alignment methods to increase performance. In addition, Busia et al. (2019) use a CNN architecture to classify 16S sequences, while other approaches also proposed to use recurrent neural networks on sequences (Ganscha et al., 2018).

More recent attempts for solving the general metagenomic classification problem focus on using deep learning to tackle it as a supervised classification task. Two examples of such attempts are *GeNet* (Rojas-Carulla et al., 2019), which attempts to leverage the hierarchical nature of taxonomic classification, and *DeepMicrobes* (Liang et al., 2019), which first learns embeddings of $k$-mers and subsequently uses those to classify each read. We use *GeNet* and a simplified version of *DeepMicrobes* as baselines and explain them in more detail in Section 3.

## 3  MODELS AND METHODS

We implemented two deep neural networks for predicting the taxa of individual reads which we use as baselines: *GeNet* (Rojas-Carulla et al., 2019) and a simplified version of *DeepMicrobes* (Liang

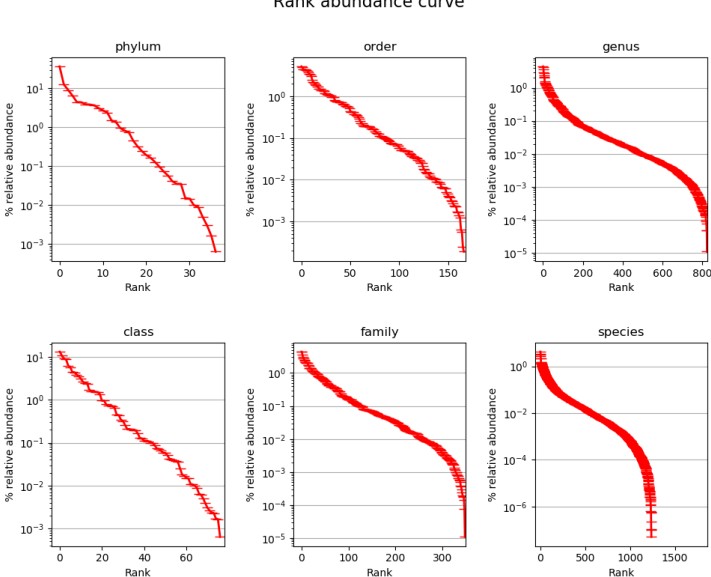

Figure 2: Rank-abundance curve for each taxonomic rank. All taxa are sorted using their abundance. Their abundance level is shown on the $y$-axis.

et al., 2019), described in sections 3.2.1 and 3.2.2 respectively. We refer to those models collectively as *single-read* models and we extend those in order to solve the MIL problem described above.

The full source code is provided online at `https://github.com/MetagenomicMIL/MetaSetMIL`.

## 3.1 DATASET GENERATION

For training, validation and evaluation, we use synthetic reads generated from bacterial genomes from the NCBI RefSeq database (Wheeler et al., 2006) from which we use a subset of 3 332 genomes comprising 1 862 species similar to the dataset used in Rojas-Carulla et al. (2019). We use NCBI's *Entrez* tool (Schuler et al., 1996), to download the genomes and the taxonomic data. The number of taxa in each taxonomic rank is summarized in Table S1 in Appendix A.

For training the single-read models, we create mini-batches in which the reads are sampled by selecting genomes uniformly at random. Training of the MIL models is different where a batch consists of a small number of bags of reads, with each bag containing reads sampled using a more realistic distribution over the genomes. The procedure used is similar to the one used by the CAMISIM simulator (Fritz et al., 2019) and described in more detail in Section 3.1.1. An example rank-abundance curve for each taxonomic rank generated by this procedure is shown in Figure 2.

From the selected genomes, we sample reads to create mini-batches in an iterative procedure similar to the procedure described in Rojas-Carulla et al. (2019). For the generation of reads, we use the software *InSilicoSeq* (Gourlé et al., 2018). We create datasets of two types in order to carry out our experiments: 151 bp reads (default length of *InSilicoSeq*) with no errors and with Illumina NovaSeq type noise. We refer to those two types of datasets as *error-free* and *novaseq*, respectively. In our experiments, we train all models on both dataset types. For validation and evaluation we only use datasets of *novaseq*-type reads in order to determine whether the models are effective at removing noise from the reads and whether it is beneficial to train with noisy reads.

Every bag is supposed to simulate a different microbial community and hence the generation procedure is repeated for each bag. The more realistic bags allow the MIL models to capture the interactions between the reads coming from related species and capture potential overlap in the reads originating from the same taxa. The validation and evaluation datasets for both single-read models and the MIL

models use this more realistic approach. Hyperparameter search was also performed for all models (details on the exact parameters can also be found in Appendix B).

### 3.1.1 SAMPLING A REALISTIC SET OF READS

In order to sample bags with a more realistic community of bacteria, we use a method similar to Fritz et al. (2019). Given a set of all the taxa $\mathcal{T}$ at a higher level (e.g., *genus* or *family*), we sample $|\mathcal{T}|$ numbers from a lognormal distribution with $\mu = 1$ and $\sigma = 2$:

$$T_i \sim Lognormal(x; \mu, \sigma) = \frac{1}{x\sigma\sqrt{2\pi}} \exp\left(-\frac{(\ln x - \mu)^2}{2\sigma^2}\right) \tag{1}$$

Then, for a taxon $t_i$ with $n$ genomes associated with it, we choose to include in our microbial community only $l_i$ random genomes where $l_i$ is sampled from a geometric distribution with $\mu = 5$:

$$P(X = l_i) = \left(1 - \frac{1}{\mu}\right)^{l_i} \frac{1}{\mu} \tag{2}$$

To calculate the abundance of a genome $g_j$ belonging to taxon $t_i$, $l_i$ random numbers $Y_1 \ldots Y_{l_i}$ are sampled from a lognormal distribution as in equation (1). The abundance for the genome is then calculated as:

$$A_j = \frac{Y_j}{\sum_{k=1}^{l_i} Y_k} T_i \tag{3}$$

All abundances are finally normalized to produce a probability vector over all the genomes in the dataset. When sampling a read set, a genome is selected by sampling from the distribution produced. Reads are then simulated from the genome sample using the software package *InSilicoSeq*.

### 3.2 BASELINE MACHINE LEARNING MODELS

#### 3.2.1 GENET

*GeNet* leverages the hierarchical nature of the taxonomy of species to simultaneously classify DNA reads at all taxonomic ranks (Rojas-Carulla et al., 2019). The procedure is similar to positional embedding as described in Gehring et al. (2017). Given an input $\boldsymbol{x} = (x_1, \ldots, x_n)$, an embedding $\boldsymbol{w} = (\boldsymbol{w}_1, \ldots, \boldsymbol{w}_n)$ is computed, where $\boldsymbol{w}_i \in \mathbb{R}^5$. The vocabulary of size 5 corresponds to the symbols for the four possible nucleotides A, C, T, G, and N (for unknown base pairs in the read). Embeddings of the absolute positions for each letter are also computed to create $\boldsymbol{p} = (\boldsymbol{p}_1, \ldots, \boldsymbol{p}_n)$, where $\boldsymbol{p}_i \in \mathbb{R}^5$. The one-hot representation of the sequence, $\boldsymbol{o}$, is added to the other two embeddings to create the matrix $\boldsymbol{w} + \boldsymbol{p} + \boldsymbol{o}$. Subsequently, the resulting matrix is passed to a ResNet-like neural network which produces a final low-dimensional representation of the read. The main novelty of the architecture is the final layer used for classification which comprises multiple softmax layers, one for each taxonomic rank. These layers are connected to each other so that information from higher ranks can be propagated towards the lower ranks. More formally, the output of softmax layer $i$ can be written as follows:

$$\boldsymbol{y}_i = ReLU(\boldsymbol{W}_i \boldsymbol{h}) + ReLU(\boldsymbol{U}_i \boldsymbol{y}_{i-1}), \tag{4}$$

where $\boldsymbol{W}_i$ and $\boldsymbol{U}_i$ are trainable parameters, $\boldsymbol{h}_i$ is the output of the ResNet network and $\boldsymbol{y}_{i-1}$ is the previous softmax output. $ReLU(\cdot)$ is the rectified linear unit function. To train the model, an averaged cross-entropy loss for each softmax layer is used.

#### 3.2.2 EMBEDPOOL

Liang et al. (2019) introduce multiple architectures for performing single-read classification among which the best is *DeepMicrobes*. It involves embedding $k$-mers ($k = 12$) into a latent representation, followed by a bidirectional LSTM, a self-attention layer, and a multi-layer perceptron (MLP). Unlike *GeNet*, this model can only be trained to classify a single taxonomic rank. Due to limited computational resources (the model requires a significant amount of GPU memory because of the very large embedding matrix), we implemented *EmbedPool*, a simpler version of *DeepMicrobes* (also

described in the original paper) to use as a baseline. In order to classify at multiple taxonomic ranks, one could run multiple instances of the model, each running on a different GPU. However, each model would be independent of the others and they would not take advantage of the hierarchical structure of the taxonomic tree. *EmbedPool* is a model that consists of an embedding layer for $k$-mers, where we set $k = 11$ in order to fit it into GPU memory. Both max- and mean-pooling are performed on the resulting matrix and concatenated together to yield a low-dimensional representation of the read. Since the embedding dimension is set to 100, after concatenation, this results in a vector of size 200. An MLP with one hidden layer of 3 000 units subsequently classifies the read. ReLU is used as the activation function. As the authors explained, most of the performance is owed to the use of the $k$-mer embedding and therefore the reduction in performance relative to *DeepMicrobes* is not expected to be significant. The model is trained end-to-end using cross-entropy loss.

## 3.3 PROPOSED MULTIPLE INSTANCE LEARNING MODELS

### 3.3.1 GENET + MIL POOLING

A mini-batch of bags of reads is used as input. The first part of *GeNet*, consisting of the embedding and the ResNet-like neural network, is used to process each read individually. A pooling layer is then used to group all reads in each bag to create bag-level embeddings. This is also referred to as MIL pooling (Foulds & Frank, 2010; Carbonneau et al., 2018). The output is passed to the final layers of *GeNet* in order to output a probability distribution over the taxa at each taxonomic rank. As a loss function we use the Jensen-Shannon ($D_{JS}$) divergence (Lin, 1991) between the predicted distribution and the actual distribution of the bag.

Given that a bag is a set, we require that a MIL pooling layer is permutation invariant, that is, permuting the reads of the bag should still produce the same result. To this end, we utilize DeepSets (Zaheer et al., 2017). DeepSets can be formally described as follows:

$$f(X) = \rho \left( \sum_{x \in X} \phi(x) \right) \tag{5}$$

In other words, each element of a set $X$ is first processed by a function $\phi(\cdot)$. The outputs are all summed together and the result is subsequently transformed by a function $\rho(\cdot)$. Zaheer et al. (2017) proved that all valid functions operating on subsets of countable sets or on fixed-sized subsets of uncountable sets can be written in this form. In our case, the inputs are embeddings in $\mathbb{R}^{5 \times L}$ where $L$ is the length of a read. In addition, we only input bags of fixed size and hence the assumptions of Theorem 2 in Zaheer et al. (2017) are satisfied. $\rho(\cdot)$ is modelled with a small MLP with one hidden layer while the ResNet part of the network models the function $\phi(\cdot)$.

Alternatively to DeepSets, we also consider an attention-based pooling layer as seen in Ilse et al. (2018) motivated by the fact that it would allow the model to attend to specific reads originating from each species. In attention-based pooling, the elements of the set are combined in different ways to create a set $\boldsymbol{z} = \boldsymbol{z}_1, \ldots, \boldsymbol{z}_k$, such that the set remains invariant when we permute the elements of the input set. This can be written as follows:

$$\boldsymbol{z}_j = \sum_{k=1}^{K} \alpha_{j,k} \boldsymbol{x}_k \ , \tag{6}$$

$$\alpha_{j,k} = \frac{\exp(\boldsymbol{w}_j^T \tanh(\boldsymbol{V} \boldsymbol{x}_k^T))}{\sum_{l=1}^{K} \exp(\boldsymbol{w}_j^T \tanh(\boldsymbol{V} \boldsymbol{x}_l^T))} \ , \tag{7}$$

where $\boldsymbol{x}_k$ is an element of the input set, and $\boldsymbol{V}$ and $\boldsymbol{w}_j$ are trainable parameters. The weights $\alpha_{j,k}$ are therefore calculated with an MLP with 1 hidden layer with $\tanh$ non-linearity and $\mathrm{softmax}$ activation at the end. Ilse et al. (2018) also attempt to increase the flexibility of the MIL pooling by introducing a gating mechanism as shown below:

$$\alpha_{j,k} = \frac{\exp(\boldsymbol{w}_j^T \left( \tanh(\boldsymbol{V} \boldsymbol{x}_k^T) \odot \sigma(\boldsymbol{U} \boldsymbol{x}_k^T) \right))}{\sum_{l=1}^{K} \exp(\boldsymbol{w}_j^T \left( \tanh(\boldsymbol{V} \boldsymbol{x}_l^T) \odot \sigma(\boldsymbol{U} \boldsymbol{x}_k^T) \right))} \ , \tag{8}$$

where $\boldsymbol{U}$ is an additional learnable matrix, $\sigma$ is the sigmoid activation function and $\odot$ is the element-wise product. As shown in Appendix B, for our models, using the gating mechanism is an additional

Table 1: Performance ($1 - D_{JS}/\ln n_t$) of all models trained on each dataset (higher is better). Our MIL models achieve superior performance at higher taxonomic ranks up to *Family*. *EmbedPool* was only trained at the *Species* level since training time exceeded our cluster limits. See subsection 4.2 for more details.

| | novaseq | | | error-free | | |
|---|---|---|---|---|---|---|
| | Phylum | Family | Species | Phylum | Family | Species |
| GeNet (Rojas-Carulla et al., 2019) | $0.871 \pm 0.017$ | $0.892 \pm 0.011$ | $0.879 \pm 0.008$ | $0.886 \pm 0.016$ | $0.895 \pm 0.011$ | $0.878 \pm 0.008$ |
| EmbedPool (Liang et al., 2019) | N/A | N/A | $\mathbf{0.903 \pm 0.009}$ | N/A | N/A | $\mathbf{0.902 \pm 0.009}$ |
| GeNet + Deepset (ours) | $\mathbf{0.985 \pm 0.008}$ | $\mathbf{0.929 \pm 0.022}$ | $0.852 \pm 0.034$ | $\mathbf{0.985 \pm 0.008}$ | $\mathbf{0.929 \pm 0.022}$ | $0.852 \pm 0.034$ |
| GeNet + Attention (ours) | $0.983 \pm 0.010$ | $0.921 \pm 0.024$ | $0.849 \pm 0.034$ | $0.984 \pm 0.008$ | $0.924 \pm 0.025$ | $0.849 \pm 0.033$ |
| Embedpool + Deepset (ours) | N/A | N/A | $0.854 \pm 0.030$ | N/A | N/A | $0.854 \pm 0.030$ |
| Embedpool + Attention (ours) | N/A | N/A | $0.853 \pm 0.033$ | N/A | N/A | $0.853 \pm 0.033$ |

hyperparameter. Following the attention mechanism, the output $z$ is flattened to create a single vector for each bag which is subsequently processed by *GeNet*'s final layers to output the predicted distributions. The overall architecture can be seen in Figure 1a.

### 3.3.2 EMBEDPOOL + MIL POOLING

Similarly to subsection 3.3.1, we use *EmbedPool* to process the reads individually. A MIL pooling layer is added after the mean- and max- pooling layers, the output of which is fed to the rest of the model to predict the distribution. JS-divergence is used as a loss function. For MIL pooling, we use DeepSets and attention-based pooling as before. An overview of the model can again be seen in Figure 1b.

## 4 RESULTS AND DISCUSSION

In this section, we analyze the results of the two baselines on solving the single-read prediction task. Then we evaluate their performance on the proposed MIL task and compare them to our MIL models. Table 1 illustrates the performance of the models trained on *novaseq* and *error-free* reads. However, in both cases the models are evaluated on *novaseq* reads in order to test their robustness to noise.

### 4.1 SINGLE-READ PREDICTIONS

In Rojas-Carulla et al. (2019), *GeNet* was trained on PacBio reads of length $10\,000$ bp and Illumina reads of length $1\,000$ bp. Since in most cases genome sequencing technologies like Illumina produce shorter reads in the range of 100 bp - 300 bp (Quail et al., 2012), we chose to train all our models on reads of length 151 bp. In the single-read prediction task *GeNet* does not perform very well on our evaluation dataset neither at *Phylum* nor *Species* levels. This is attributed to the fact that it might be unable to extract useful features shared across the whole genome from shorter reads, especially because one-hot encoding is used rather than $k$-mer encoding. On the other hand, even though *EmbedPool* seemed to be performing well during training, achieving training accuracy of $0.789$, when the distribution of the reads in the mini-batch is changed (as is the case with our more realistic evaluation dataset), the accuracy drops to $0.223$. This signifies that *EmbedPool* is not able to accurately classify all species equally well. *GeNet* however seems to be more robust to the change of the mini-batch distribution since the accuracy does not drop when moving from the training dataset to the more realistic evaluation dataset. In addition, training with noisy reads seems to not improve results for *EmbedPool* when evaluating the classifier on noisy reads. However, training with *error-free* reads seems to achieve better results for *GeNet* even when evaluating on *novaseq* reads. A table with the accuracy achieved by both baselines in the single-read prediction task can be found in Appendix A.

### 4.2 READ-SET-BASED PREDICTIONS

In each taxonomic rank $t$, the upper bound for the JS-divergence differs because of different numbers of taxa $n_t$ belonging to that rank. Therefore, we normalize our results and use $1 - D_{JS}/\ln n_t$ as the metric for comparison, where a value of $1.0$ means the model achieved perfect performance. Table 1 shows a comparison of our MIL models and the achieved scores. A table of the raw $D_{JS}$ values can

be found in Appendix A. For the standard *GeNet* and *EmbedPool*, the microbiota distribution was calculated by classifying each read independently while for the rest, the distribution was predicted directly by the models. An example of the output of the MIL models is shown in Figure 3. All models were evaluated on a total of 100 bags of 2048 *novaseq*-type reads each. Both *GeNet + Deepset* and *GeNet + Attention* perform better than standard *GeNet* at higher taxonomic ranks. As explained in Section 1, we believe that the improvement in accuracy is owed to the fact that the models can exploit the co-occurrence of species in realistic settings or detect overlaps of reads in a bag. A drawback of our MIL models is that, since the performance is owed to the special structure of the bags, it is unlikely that they would perform well when presented with bags with an unrealistic distribution of species (e.g., a bag with a uniformly random distribution over all species). Therefore, it is clear that the models achieve a trade-off between flexibility and performance. Moreover, our proposed MIL models perform poorly on the finer taxonomic ranks, possibly because in the MIL setting, the models only observe a summary of the bag rather than a label for each instance and it is therefore harder for them to learn adequate features. However, the greater performance on higher levels can prove beneficial for some real-world metagenomic datasets where sufficient reference data is not available to train deep learning models accurately (Afshinnekoo et al., 2015; Tully et al., 2018). A comparison of *GeNet + Deepset*, our best performing model and standard *GeNet* can be seen in Figure S1 in Appendix A.

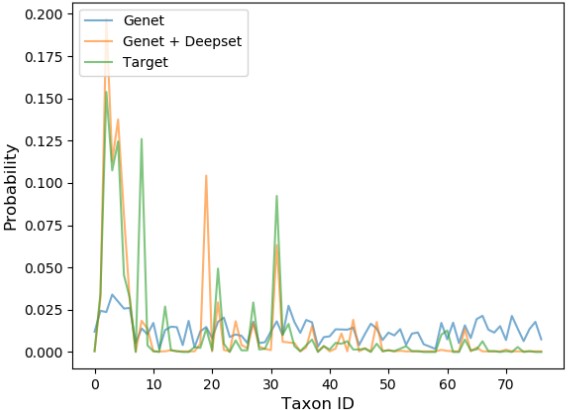

Figure 3: Distribution of taxa at the class rank. The target distribution is denoted in orange and the output of the model is denoted in blue.

## 5 CONCLUSIONS

In this work, we tackle the problem of directly predicting the distribution of the microbiota in metagenomic samples. In contrast to previous methods that are based on classifying single reads, we formulate the problem as a Multiple Instance Learning task and use permutation invariant pooling layers in order to learn low-dimensional embeddings for whole sets of reads. We show that our proposed method can perform better than the baseline models at the higher taxonomic ranks. The MIL models presented could be used as an initial step to filter or preselect the potential genomes that more traditional alignment methods would need to take as input in order to increase their performance.

Further work could include exploring alternative base architectures or more sophisticated pooling methods that can better capture the interactions between reads. For example, one could use Janossy pooling (Murphy et al., 2018), another permutation invariant method that can capture $k$-order interactions between the elements of a set. Also, the models could potentially be combined with a probabilistic component, such as a Gaussian process over DNA sequences (Fortuin et al., 2018), to allow for uncertainty estimates on the predictions. Finally, as explained, a possible issue is that observing only the summary of the read set can make it more difficult for the model to learn adequate features for the individual reads. A solution to this could be to first learn better instance-level embeddings to use as input, in order to aid the model in learning suitable bag-level embeddings.

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

## A  SUPPLEMENTARY MATERIAL

To train and test our models, we have downloaded 3 332 genomes from the NCBI RefSeq database (Wheeler et al., 2006). The full list of accession numbers for the genomes used in our dataset can be found in our GitHub repository (`https://github.com/MetagenomicMIL/MetaSetMIL`).

Table S1: Number of taxa per rank in our dataset. The selected accession numbers are a subset of the dataset used by Rojas-Carulla et al. (2019). See subsection 3.1

| Rank | # of taxa |
|---|---|
| Phylum | 37 |
| Class | 77 |
| Order | 167 |
| Family | 349 |
| Genus | 824 |
| Species | 1862 |

The accuracy of the two baseline models at solving the single-read prediction task was evaluated and the results are shown in Table S2.

Table S2: Accuracy of the two base models trained on each dataset (higher is better).

| | novaseq | | error-free | |
|---|---|---|---|---|
| | Phylum | Species | Phylum | Species |
| GeNet | $0.258 \pm 0.032$ | $0.100 \pm 0.015$ | $0.290 \pm 0.034$ | $0.097 \pm 0.017$ |
| EmbedPool | N/A | $0.223 \pm 0.028$ | N/A | $0.224 \pm 0.030$ |

Subsequently, all models were evaluated on solving the MIL task. The JS-divergence achieved by all models is shown in Table S3 while a comparison of our best performing model, *GeNet + Deepset*, and *GeNet* is depicted in Figure S1.

Table S3: JS-divergence for all models trained on each dataset. Our MIL models achieve superior performance at higher taxonomic ranks up to *Family*. See subsection 4.2 for more details.

| | novaseq | | | error-free | | |
|---|---|---|---|---|---|---|
| | Phylum | Family | Species | Phylum | Family | Species |
| GeNet (Rojas-Carulla et al., 2019) | $0.466 \pm 0.062$ | $0.633 \pm 0.064$ | $0.912 \pm 0.057$ | $0.412 \pm 0.057$ | $0.614 \pm 0.063$ | $0.920 \pm 0.059$ |
| EmbedPool (Liang et al., 2019) | N/A | N/A | $\mathbf{0.733 \pm 0.064}$ | N/A | N/A | $\mathbf{0.741 \pm 0.067}$ |
| GeNet + Deepset (ours) | $\mathbf{0.053 \pm 0.028}$ | $\mathbf{0.417 \pm 0.131}$ | $1.115 \pm 0.253$ | $\mathbf{0.053 \pm 0.029}$ | $\mathbf{0.416 \pm 0.131}$ | $1.115 \pm 0.253$ |
| GeNet + Attention (ours) | $0.062 \pm 0.035$ | $0.462 \pm 0.139$ | $1.135 \pm 0.253$ | $0.058 \pm 0.030$ | $0.446 \pm 0.145$ | $1.140 \pm 0.251$ |
| Embedpool + Deepset (ours) | N/A | N/A | $1.101 \pm 0.228$ | N/A | N/A | $1.101 \pm 0.227$ |
| Embedpool + Attention (ours) | N/A | N/A | $1.107 \pm 0.247$ | N/A | N/A | $1.105 \pm 0.245$ |

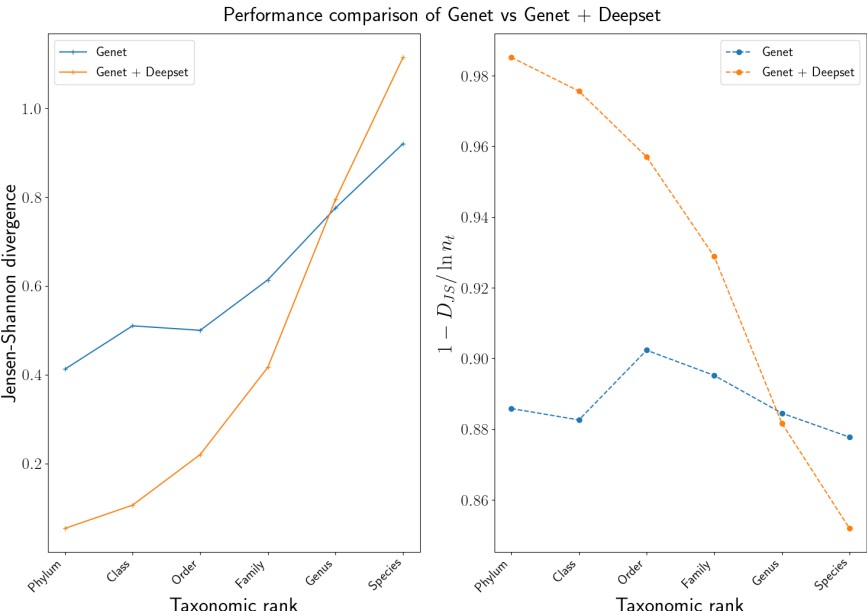

Figure S1: Performance comparison of *GeNet* vs *GeNet + Deepset*. *GeNet + Deepset* achieves superior performance on taxonomic ranks upto *Family*.

# B  HYPERPARAMETER GRID FOR THE TRAINED MODELS.

To train our models, we performed random search over the following hyperparameter grid:

Table S4: Hyperparameter grid

| General parameters for single read models | |
|---|---|
| Batch Size | 64, 128, 256, 512, 1024, 2048 |
| **General parameters for MIL models** | |
| Bag Size | 64, 128, 512, 1024, 2048 |
| Batch Size | 1, 2, 4, 8 |
| **GeNet** | |
| Output size of ResNet | 128, 256, 512, 1024 |
| Use GeNet initialization scheme | True, False |
| BatchNorm running statistics | True, False |
| Optimizer | Adam, SGD |
| Learning rate | 0.001, 0.0005, 1.0 (for SGD) |
| Nesterov momentum (SGD only) | 0.0, 0.9, 0.99 |
| **EmbedPool** | |
| Size of MLP hidden layer | 1000, 3000 |
| Optimizer | Adam, RMSprop, SGD |
| Nesterov momentum (SGD only) | 0.0, 0.5, 0.9, 0.99 |
| Learning rate | 0.001, 0.0005 |
| **Deepset pooling layer** | |
| Deepset $\rho$ hidden layer size | 128, 256, 1024 |
| Deepset output size | 128, 1024 |
| Dropout before $\rho$ network | 0.0, 0.2, 0.5, 0.8 |
| Deepset activation | ReLU, Tanh, ELU |
| **Attention pooling layer** | |
| Hidden layer size | 128, 256, 512, 1024 |
| Gated attention | False, True |
| Attention rows | 1, 10, 30, 60 |

