# OpenReview forum: "Deep Multiple Instance Learning for Taxonomic Classification of Metagenomic read sets"
_ICLR.cc/2020/Conference — Reject_

### Official Review · AnonReviewer1 · 2019-10-20
**Official Blind Review #1**

**Rating:** 3

**Review:**

The authors tackle the task of taxonomic classification of meta genomic read sets. The combine 2 existing approaches for taxonomic classification, with established methods for Multiple Instance Learning, namely DeepSets and an attention-based pooling layer.

While the domain of taxonomic classification is interesting, I find there is a lack of novelty on the machine learning part. The authors combine well established methods in a straight-forward manner and while the resulting increase in performance for some datasets may be relevant in the domain, the conceptual advances are too incremental for a machine learning audience.

Update: I have read the response and am still think there is a lack of novelty here.

**Experience Assessment:**

I have read many papers in this area.

**Review Assessment: Checking Correctness Of Derivations And Theory:**

I carefully checked the derivations and theory.

**Review Assessment: Checking Correctness Of Experiments:**

I carefully checked the experiments.

**Review Assessment: Thoroughness In Paper Reading:**

I read the paper thoroughly.

---

> ### Author Response · Authors · 2019-11-14
> **Reply to Reviewer #1**
>
> Thank you for taking the time to review our paper.
>
> We believe the novelty on the machine learning part is using Multiple Instance Learning (MIL) to predict a distribution over a bag of instances. Most papers on MIL in both classification and regression settings focus on bags where only some instances (called the primary instances) contribute to the label/value of the bag and there is a clear distinction between positive and negative instances [1]. To our knowledge, no paper has used MIL combined with Jensen-Shannon divergence as a loss function, for predicting distributions where all instances are likely to contribute to the final result. The increase in performance demonstrates that there is global information in the structure of a bag, that can be exploited to increase the predictive power of models, which is lost when instances are processed individually instead.
>
> [1] Carbonneau, M.A., Cheplygina, V., Granger, E. and Gagnon, G., 2018. Multiple instance learning: A survey of problem characteristics and applications. Pattern Recognition, 77, pp.329-353.

---

### Official Review · AnonReviewer3 · 2019-10-23
**Official Blind Review #3**

**Rating:** 1

**Review:**

~The authors propose the addition of multiple instance learning mechanism to existing deep learning models to predict taxonomic labels from metagenomic sequences.~

I appreciate the focus area and importance of the problem the authors have outlined. However, I do not think the authors have achieved the conclusions they mention on page 2, as well as other issues throughout the work. I also think the inclusion of the multiple instance learning framework is incremental and does not provide sufficient benefit.

“A new method to generate synthetic read sets with realistic co-occurence patterns from collections of reference genomes”. I do not think there was any systematic analysis of the parameterization of their generative framework. I would appreciate empirical comparison of previous synthetic read generation techniques to the current proposed framework. Also, there is no comparison of the generative framework to real data. How are the parameters chosen in section 3.1.1? Finally, how the authors propose to alleviate bias of composition of databases? Rare species that may be present in abundance in metagenomic data may be swamped out by more common species sequenced again and again in databases.

“A thorough empirical assessment of our proposed model, showing superior performance in prediction the distributions of higher level taxa from read sets.” A comparison to existing alignment-based methods is absolutely required for this work. The authors of GeNet compare to state-of-art Kraken and Centrifuge, and when reading Rojas-Carulla et al., these models have still performed worse than Kraken and Centrifuge, and that should be reported in your assessment.

A few minor points:

-More description of your neural network architecture is needed. I’m not sure what a “ResNet-like neural network” actually means, and how something built for images deals with sequences. There are also different ResNets with different numbers of parameters.

-Why isn’t a 1D convolutional neural network used to process the input sequences? This would make the sequences translation invariant, and would have a similar effect to working with kmers, where k = convolution width.

-It may be useful to interpret the attention mechanism to understand which reads are likely influencing the decision of taxonomic assignment.


**Experience Assessment:**

I have published in this field for several years.

**Review Assessment: Checking Correctness Of Derivations And Theory:**

I carefully checked the derivations and theory.

**Review Assessment: Checking Correctness Of Experiments:**

I carefully checked the experiments.

**Review Assessment: Thoroughness In Paper Reading:**

I read the paper thoroughly.

---

> ### Author Response · Authors · 2019-11-14
> **Reply to Reviewer #3**
>
> Thank you for taking the time to review our paper. We would like to address some of the issues raised.
>
> "I also think the inclusion of the multiple instance learning framework is incremental and does not provide sufficient benefit."
> We believe the novelty on the machine learning part is using Multiple Instance Learning (MIL) to predict a distribution over a bag of instances. Most papers on MIL in both classification and regression settings focus on bags where only some instances (called the primary instances) contribute to the label/value of the bag and there is a clear distinction between positive and negative instances [1]. To our knowledge, no paper has used MIL combined with Jensen-Shannon divergence as a loss function, for predicting distributions where all instances are likely to contribute to the final result. The increase in performance demonstrates that there is global information in the structure of a bag, that can be exploited to increase the predictive power of models, which is lost when instances are processed individually instead.
>
> “I do not think there was any systematic analysis of the parameterization of their generative framework”
> The parameters used are the default values from [2]. The parameters can be modified depending on the type of microbial community being simulated. The idea was to just insert some co-occurence patterns into our simulated dataset to test our idea of exploiting the global structure in the bag in order to increase the model performance.
>
> “A comparison to existing alignment-based methods is absolutely required for this work”
> We agree that the inclusion of traditional-alignment based baselines such as Kraken and Centrifuge is very important. Given the limited space, we made the decision to not include traditional baselines in the manuscript and considered including state-of-the-art ML methods more important. However, we do plan to include a comparison with traditional methods in a future version of our manuscript.
>
> Minor Points:
> Generally, CNNs have been used for text classification before, for example in [3, 4]. As explained in the manuscript, by 'ResNet-like neural network', we refer to the neural net as described by Rojas-Carulla et al. and it is described in more detail in [5]. In addition, anonymized code has been provided which shows our implementation of the particular ResNet-like network. Indeed, a 1D kernel is used as part of the network.
>
> Again, thank you for your feedback and if you have any further suggestions, please let us know.
>
>
> [1] Carbonneau, M.A., Cheplygina, V., Granger, E. and Gagnon, G., 2018. Multiple instance learning: A survey of problem characteristics and applications. Pattern Recognition, 77, pp.329-353.
>
> [2] Fritz, A., Hofmann, P., Majda, S., Dahms, E., Dröge, J., Fiedler, J., Lesker, T.R., Belmann, P., DeMaere, M.Z., Darling, A.E. and Sczyrba, A., 2019. CAMISIM: simulating metagenomes and microbial communities. Microbiome, 7(1), p.17.
>
> [3] Conneau, A., Schwenk, H., Barrault, L. and Lecun, Y., 2016. Very deep convolutional networks for text classification. arXiv preprint arXiv:1606.01781.
>
> [4] Kim, Y., 2014. Convolutional neural networks for sentence classification. arXiv preprint arXiv:1408.5882.
>
> [5] Rojas-Carulla, M., Tolstikhin, I., Luque, G., Youngblut, N., Ley, R. and Schölkopf, B., 2019. GeNet: Deep Representations for Metagenomics. arXiv preprint arXiv:1901.11015.

---

### Official Review · AnonReviewer2 · 2019-10-24
**Official Blind Review #2**

**Rating:** 3

**Review:**

I enjoyed reading this paper and found much of their exposition clear. Also found their extension of previous single read metagenomic classification models with DeepSets and attentional pooling layers to be well explained. However, there are two significant flaws that unfortunately make this paper incomplete as written:

1) The abstract states that this paper will "attempt to solve the task of directly predicting the distribution over the taxa of whole metagenomic read sets". However, "whole metagenomic read sets" typically contain many millions of reads, whereas the maximum bag size explored in this paper is 2048.  The authors never explain how their MIL metagenomic model should be applied to a full metagenomic sequencing dataset. Should the MIL model be applied to random 2048 read subsets of the many-million read real world datasets? Should these bags of reads be sampled with or without replacement? And, when applied to a whole metagenomic read set, are the improvements in classification accuracy observed for 2048 read bags recapitulated?

2) There is no comparison to standard metagenomic classification tools such as Kraken and Centrifuge. While previous work such as GeNet have compared to these tools the read generation and testing assumptions in this paper are not identical. Also, GeNet was found to be inferior to Kraken and Centrifuge in many scenarios, it would be good to know where in the spectrum of accuracy these new models fall.

Update: Having read the rebuttal, my review stands. This paper will be much better once the authors add in estimation from full read sets and evaluate against standard tools. Presumably that will have to be for a future conference submission.

**Experience Assessment:**

I have read many papers in this area.

**Review Assessment: Checking Correctness Of Derivations And Theory:**

I assessed the sensibility of the derivations and theory.

**Review Assessment: Checking Correctness Of Experiments:**

I assessed the sensibility of the experiments.

**Review Assessment: Thoroughness In Paper Reading:**

I read the paper at least twice and used my best judgement in assessing the paper.

---

> ### Author Response · Authors · 2019-11-14
> **Reply to Reviewer #2**
>
> Thank you for taking the time to review our paper and thank you for your comments.
>
> “The authors never explain how their MIL metagenomic model should be applied to a full metagenomic sequencing dataset”
> Sampling is not required for this method to scale. In the simpler case of DeepSets, the sum can be decomposed to a sum of sums. In other words, we can process 2048 reads at a time up to and including the MIL pooling phase and 'pause' processing there in order to get a representation for each subset of reads. We can then sum all the representations together to get a vector representation for the whole readset and continue with the execution of the rest of the network. Since we can accumulate the representations into a single vector instead of storing each individual subset representation, we can avoid any potential GPU memory issues we would have with a much larger readset.
> A similar, albeit less efficient, decomposition can be done for the attention-based pooling. For example, one could first calculate the numerator \exp(w^T \tanh(V x_k^T)) (Eq. 7 in the paper) for each read (processing 2048 reads a time). All the numerators need to be stored since these values need to be normalized to be between 0 and 1. Storing the numerators can be done in a secondary GPU or RAM where no other parameters of the model are stored. Normalization is then done in the secondary GPU and a single vector representing the weighted average can be transferred back to the main GPU for further processing by the remaining layers of the network. Following this approach, no additional randomness or noise is introduced to the model through sampling with/without replacement.
> We do agree however, that this idea could be more thoroughly tested with additional experiments.
>
> “There is no comparison to standard metagenomic classification tools such as Kraken and Centrifuge”
> We agree that the inclusion of traditional alignment-based baselines such as Kraken and Centrifuge is very important. Given the limited space, we made the decision to not include traditional baselines in the manuscript and considered including state-of-the-art ML methods more important. However, we do plan to include a comparison with traditional methods in a future version of our manuscript.
>
> We will take your feedback into account for future revisions of our paper. Again, thank you for your review and if you have any further suggestions, please let us know.

---

### Decision · Program_Chairs · 2019-12-19

**Decision:**

Reject

**Comment:**

The work proposes a modification to existing architectures applied to predict taxonomic labels from metagenomic sequences. Reviewers agreed that the problem was well motivated, but that current experiments lack comparisons with existing standard baselines in the area. I recommend the authors update their work to included the additional experiments suggested by the reviewers.